# Landscapes of Enteric Virome Signatures in Early-Weaned Piglets

Shiyu Tao,[a] Huicong Zou,[a] Jingjing Li,[a] Hong Wei[a]

aCollege of Animal Sciences and Technology, Huazhong Agricultural University, Wuhan, China

**ABSTRACT** Diarrhea caused by early-weaning-induced stress can increase mortality rates and reduce growth performance of piglets, seriously harming the livestock industry. To date, studies on the gut microbiome of early-weaned piglets have focused almost exclusively on bacteria, while studies on their gut virome are extremely lacking. Here, we used metagenomic and metatranscriptomic sequencing combined with bioinformatic analysis techniques to preliminarily characterize the intestinal virome of early-weaned piglets at different biological classification levels. The alpha diversity of enteroviruses was generally elevated in early-weaned piglets with diarrhea, compared to healthy piglets, whereas the two groups of piglets showed no significant difference in beta diversity. In addition, the species compositions of the gut virome were similar between healthy piglets and piglets with diarrhea, while their respective dominant species were somewhat different. We also identified 58 differential DNA viruses and 16 differential RNA viruses between the two groups of piglets at all biological taxonomic levels. Of these, 1 (family *Dhakavirus*) and 6 (phylum *Artverviricota*, class *Revtraviricetes*, order *Ortervirales*, family *Retroviridae*, genus *Gammaretrovirus*, and species Kirsten murine sarcoma virus) specific viruses disappeared from the intestines of healthy piglets and piglets with diarrhea, respectively. Moreover, we found that some DNA and RNA viruses formed strong correlations among themselves or between them.

**IMPORTANCE** This study systematically reveals the biological diversity, structure, and composition of intestinal DNA and RNA virus profiles in early-weaned piglets. Furthermore, characteristics of differences in gut viromes between early-weaned healthy piglets and piglets with diarrhea were also elucidated. Importantly, some potential biomarkers for early-weaned piglets with diarrhea were identified. These findings fill a gap for the early-weaned piglet gut virome and lay the foundation for the development of strategies to target enteroviruses for the prevention and treatment of early-weaning-induced piglet diarrhea.

**KEYWORDS** early weaning, gut virome, metagenome, metatranscriptome, piglet

Weaning is a key stage that pigs must go through in the process of the growth cycle. Due to the change of diet and psychological and environmental conditions, weaned piglets may experience rapid stress and other uncomfortable reactions, which then lead to the occurrence of symptoms such as decreased appetite, poor digestion, intestinal inflammation, and intestinal microbiota disorders, which seriously affect the growth and healthy of piglets (1, 2). The natural weaning time of piglets is usually about 17 weeks after birth; however, early-weaning measures at 21 days of age are commonly used in intensive pig farms (3). Early weaning helps to improve the reproductive performance of sows by shortening the estrus interval, shortens the slaughter time for pigs by promoting growth throughout the life cycle, improves the utilization of the pen, and reduces the risk of disease transmission from sows to piglets, thus significantly improving farming efficiency (4). However, early weaning can further stimulate piglets, exacerbate weaning stress, and lead to impaired intestinal epithelial barrier function, greatly increasing the risk of intestinal dysfunction in piglets (5–7). In addition, early weaning leads to electrolyte imbalances in

Address correspondence to Shiyu Tao, sytao@mail.hzau.edu.cn, or Hong Wei, weihong63528@163.com.

The authors declare no conflict of interest.

piglets' intestine, decreased expression of intestinal fluid absorption proteins in intestinal epithelial cells, and increased expression of intestinal fluid secretion proteins, causing a surge of intestinal fluid in the intestinal lumen and into the hindgut, which eventually leads to diarrhea and even death in piglets (8).

The intestinal microbiome includes bacteria, viruses, fungi, and archaea. Currently, the vast majority of studies targeting the microbiome have focused on bacteria (9–13). However, the important role played by the virome, a complex community of eukaryotic RNA viruses, DNA viruses, and bacteriophages, in maintaining animal healthy is often overlooked (14). The intestinal virome has a nonnegligible role in maintaining the intestinal microecosystems and is an important component of the intestinal microbiome independent of the intestinal bacteria, although the two represent a unified organism (15–17). Gut virome sequencing and analysis are emerging technologies that have evolved from metagenomics to study viruses, which can fully explore the viral communities in specific environments and have been widely used in humans (18, 19). With the availability of high-throughput sequencing technologies, researchers are increasingly studying the structure and function of the intestinal microbiota of early-weaned piglets (20–22), and those studies have greatly improved the scientific community's knowledge of the characteristics of the intestinal microbiota of early-weaned piglets. However, no reports on the intestinal viral profile of early-weaned piglets have been found.

Here, we examined DNA and RNA viruses in the feces of early-weaned healthy piglets and piglets with diarrhea using metagenomic and metatranscriptomic techniques, respectively. We further performed bioinformatic analysis of DNA and RNA viral profiles obtained from sequencing to reveal the structural and compositional characteristics of the viral communities in early-weaned piglets with different biological classifications. The differentiation of the gut virome was also elucidated in piglets with different health conditions after early weaning. Our study is the first to systematically analyze the landscape of intestinal viral signatures of early-weaned piglets, which will help fill the knowledge gap in the field and lay the foundation for more in-depth studies in the future.

## RESULTS

**Alpha diversity of fecal DNA and RNA viruses in early-weaned healthy piglets and piglets with diarrhea.** Initially, to explore whether there is a difference in the alpha diversity of intestinal DNA and RNA viruses in early-weaned healthy piglets and piglets with diarrhea, we analyzed richness, Shannon, and Simpson indexes at different biological classification levels (phylum, class, order, family, genus, and species). For DNA viruses, the richness index of early-weaned piglets with diarrhea was significantly higher than that of healthy piglets at the family and species levels ($P < 0.05$) (Fig. 1A), while the richness index of the two groups of piglets was not significantly different at the phylum, class, order, and genus levels ($P > 0.05$) (Fig. 1A). In addition, the Shannon and Simpson indexes for early-weaned healthy piglets and piglets with diarrhea did not differ at any biological classification level ($P > 0.05$) (Fig. 1B and C).

For RNA viruses, the richness index of early-weaned piglets with diarrhea was significantly greater than that of healthy piglets at the genus and species levels ($P < 0.05$) (Fig. 2A), while the richness indexes of the two groups of piglets were not significantly different at the phylum, class, order, and family levels ($P > 0.05$) (Fig. 2A). Compared to early-weaned healthy piglets, the Shannon index was significantly higher in piglets with diarrhea at the phylum, class, order, family, and genus levels ($P < 0.05$) (Fig. 2B), while there was no change at the species level ($P > 0.05$) (Fig. 2B). Moreover, the Simpson index was significantly increased in early-weaned piglets with diarrhea at the phylum and family levels ($P < 0.05$) (Fig. 2C), while there was no change at other biological classification levels ($P > 0.05$) (Fig. 2C).

**Beta diversity of fecal DNA and RNA viruses in early-weaned healthy piglets and piglets with diarrhea.** Then, we sought to explore whether the overall viral phenotypes of early-weaned healthy piglets and piglets with diarrhea were different.

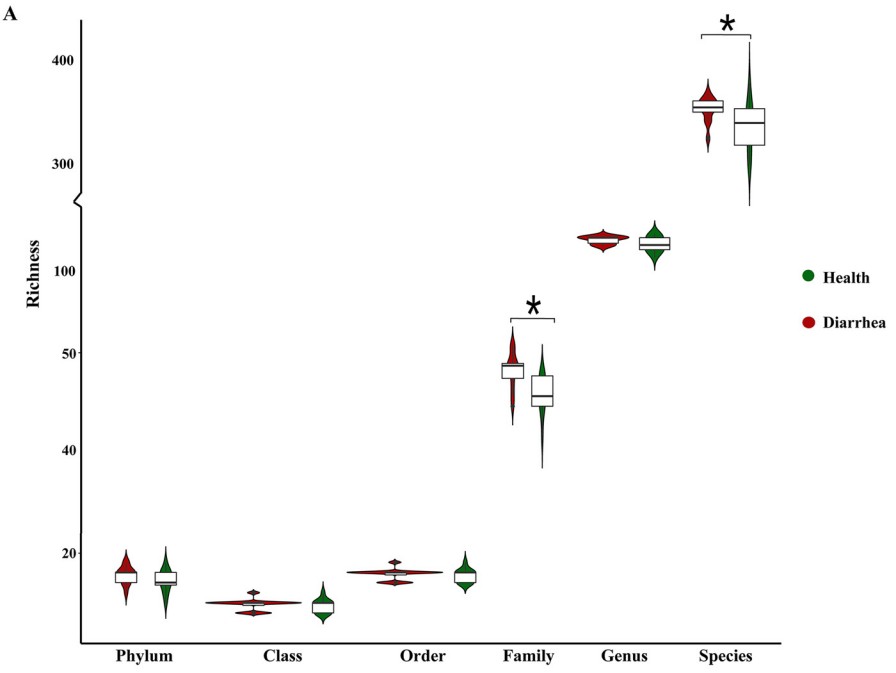

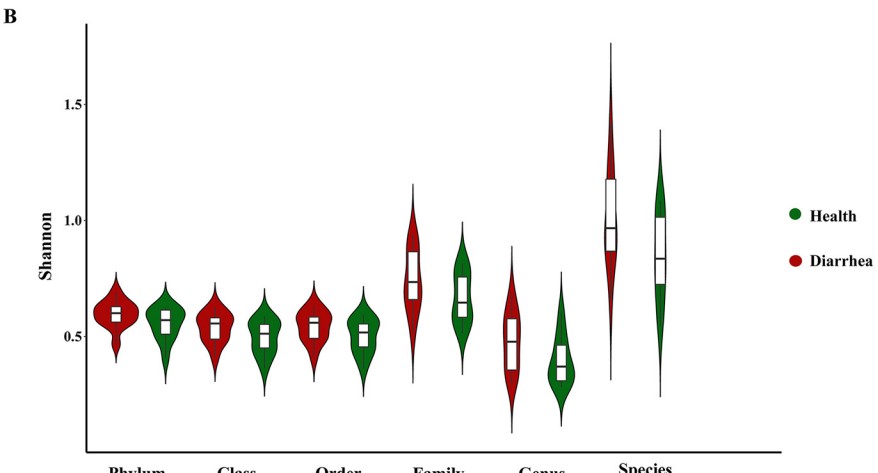

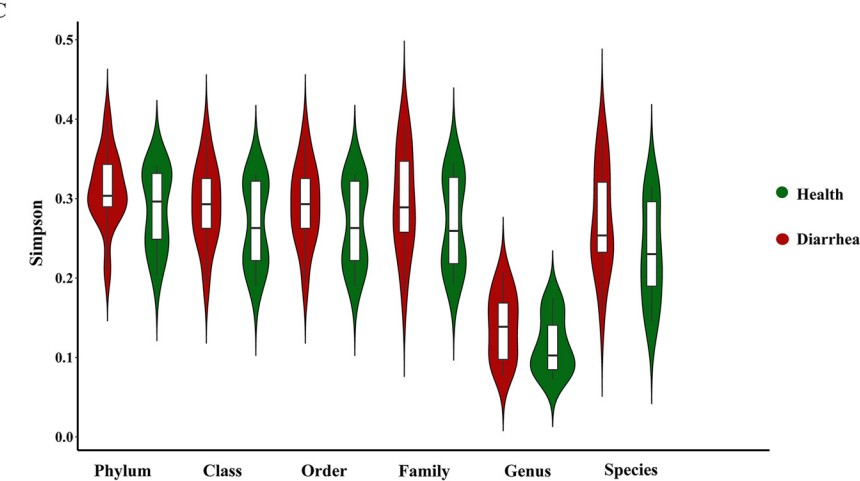

**FIG 1** Violin plots of alpha diversity of intestinal DNA viruses in early-weaned healthy piglets and piglets with diarrhea. Richness (A), Shannon (B), and Simpson (C) indexes at the phylum, class, order, family, genus, and species levels are shown. *, $P < 0.05$ ($n = 12$).

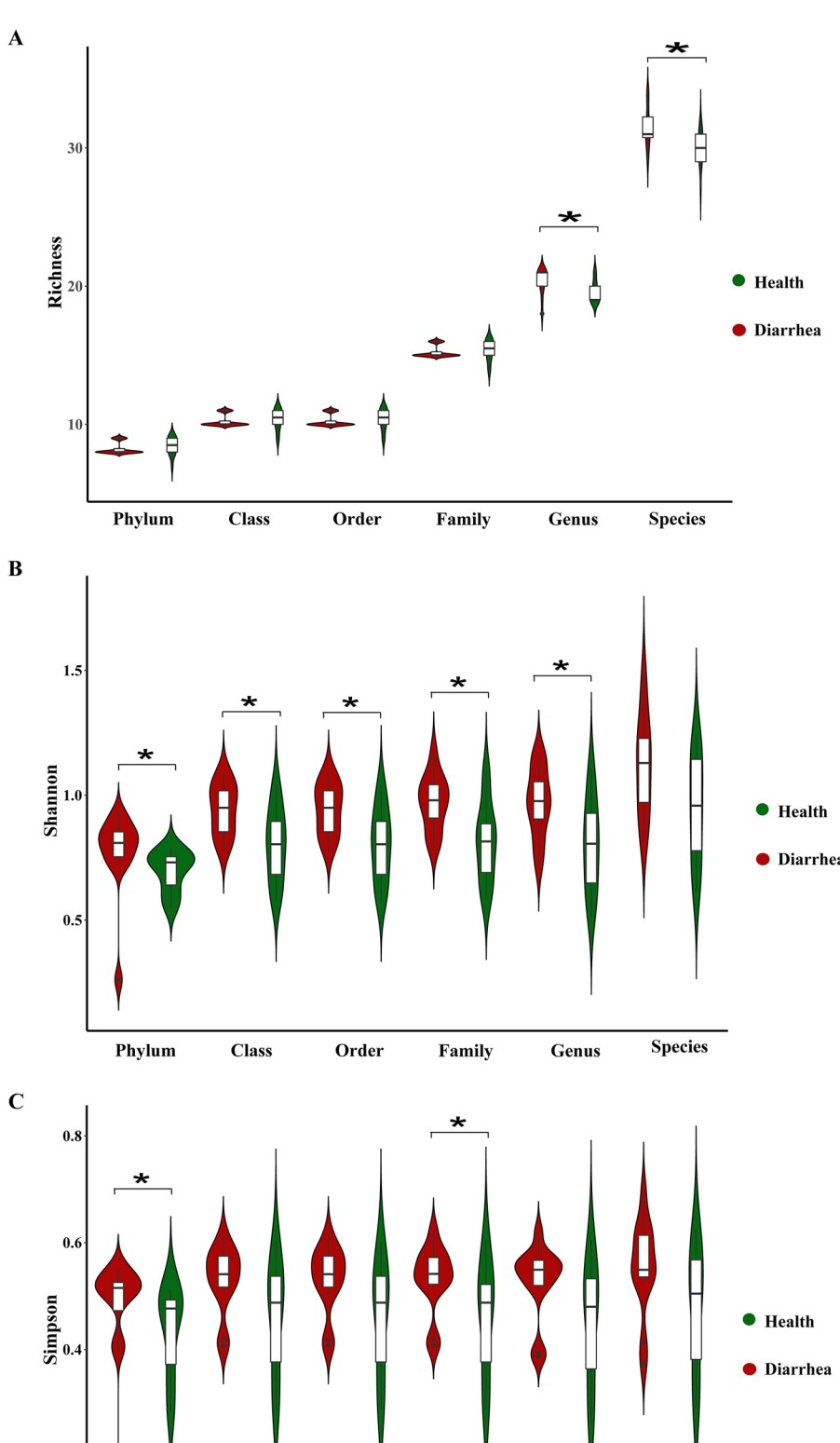

**FIG 2** Violin plots of alpha diversity of intestinal RNA viruses in early-weaned healthy piglets and piglets with diarrhea. Richness (A), Shannon (B), and Simpson (C) indexes at the phylum, class, order, family, genus, and species levels are shown. *, $P < 0.05$ ($n = 12$).

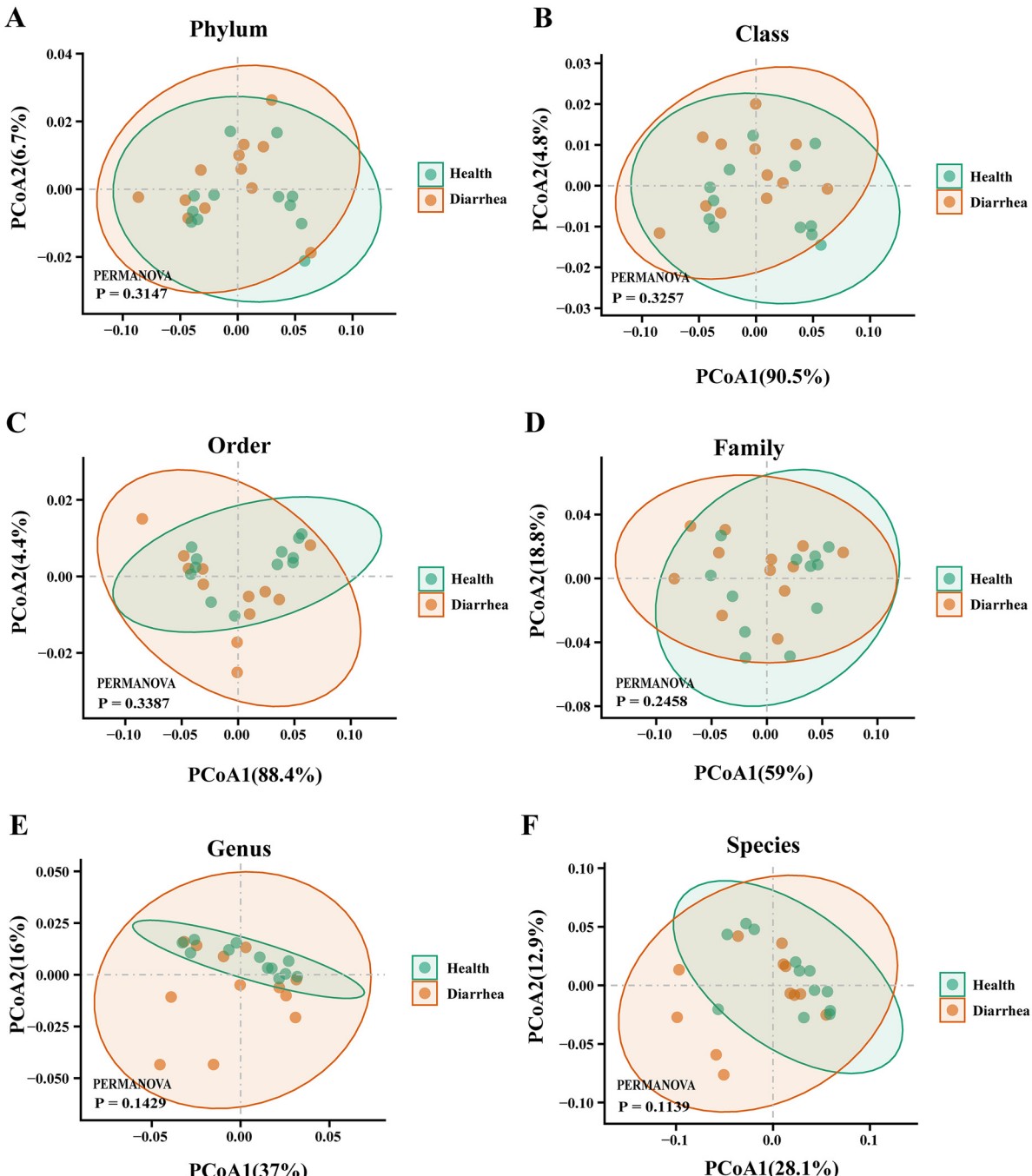

**FIG 3** PCoA plots of beta diversity of intestinal DNA viruses in early-weaned healthy piglets and piglets with diarrhea at the phylum (A), class (B), order (C), family (D), genus (E), and species (F) levels.

Principal-coordinate analysis (PCoA) showed that DNA viral signatures for the two groups were not significantly distinct at all biological classification levels ($P > 0.05$) (Fig. 3A to F). Moreover, the PCoA results for RNA viruses in early-weaned healthy piglets and piglets with diarrhea were similar ($P > 0.05$) (Fig. 4A to F).

**Species composition of fecal DNA and RNA viruses in early-weaned healthy piglets and piglets with diarrhea.** The species compositions of DNA and RNA viruses from weaned piglets in all health conditions at all biological taxonomic levels are shown in Fig. 5 and 6. For DNA viruses at the phylum level, *Uroviricota*, *Nucleocytoviricota*, and *Peploviricota* were the predominant viruses in feces of healthy piglets, whereas *Uroviricota*,

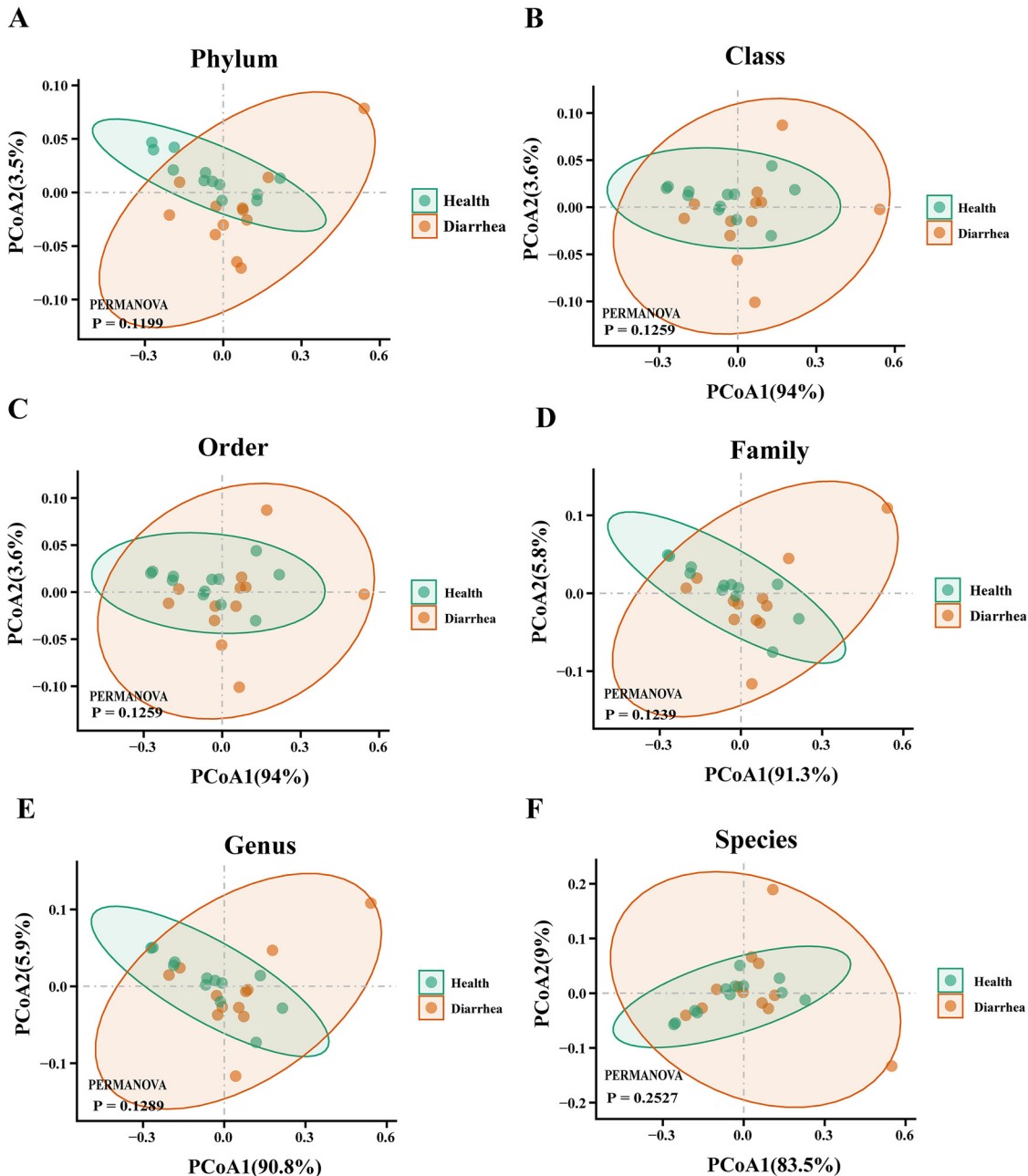

**FIG 4** PCoA plots of beta diversity of intestinal RNA viruses in early-weaned healthy piglets and piglets with diarrhea at the phylum (A), class (B), order (C), family (D), genus (E), and species (F) levels.

*Nucleocytoviricota*, and *Negarnaviricota* were the predominant viruses in feces of piglets with diarrhea (Fig. 5A). At the class level, *Caudoviricetes*, *Megaviricetes*, and *Herviviricetes* were the predominant viruses in feces of healthy piglets, whereas *Caudoviricetes*, *Megaviricetes*, and *Ellioviricetes* were the predominant viruses in feces of piglets with diarrhea (Fig. 5B). At the order level, *Caudovirales*, *Algavirales*, and *Herpesvirales* were the predominant viruses in feces of healthy piglets, whereas *Caudovirales*, *Pimascovirales*, and *Imitervirales* were the predominant viruses in feces of piglets with diarrhea (Fig. 5C). At the family level, *Siphoviridae*, *Podoviridae*, and *Myoviridae* were the predominant viruses in feces of healthy piglets, whereas *Podoviridae*, *Siphoviridae*, and *Myoviridae* were the predominant viruses in feces of piglets with diarrhea (Fig. 5D). At the genus level, *Oengusvirus*, *Phikzvirus*, and

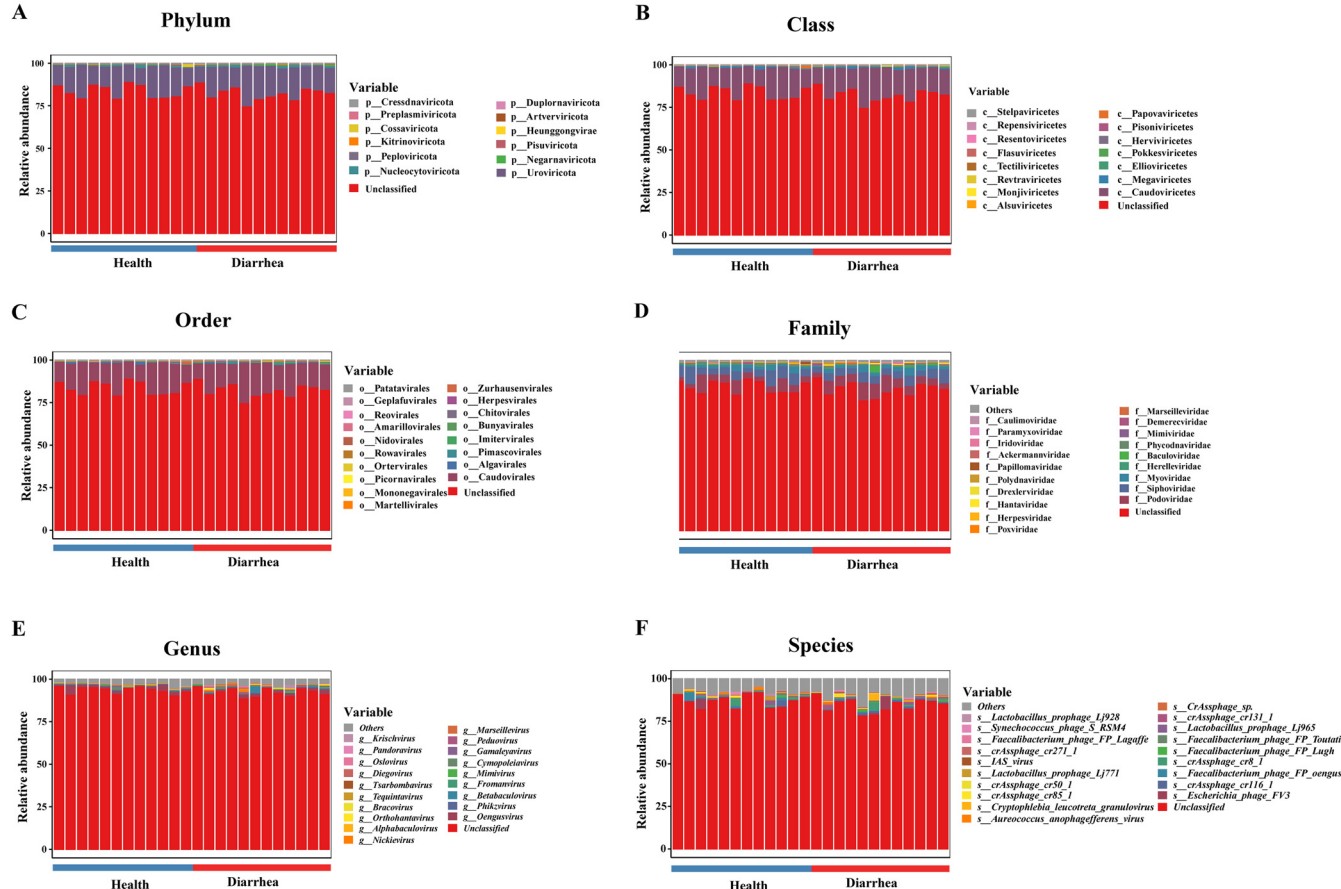

**FIG 5** Composition of intestinal DNA viruses in early-weaned healthy piglets and piglets with diarrhea at the phylum (A), class (B), order (C), family (D), genus (E), and species (F) levels.

*Fromanvirus* were the predominant viruses in feces of healthy piglets, whereas *Oengusvirus*, *Betabaculovirus*, and *Peduovirus* were the predominant viruses in feces of piglets with diarrhea (Fig. 5E). At the species level, *Escherichia* phage FV3, crAssphage cr116_1, and *Faecalibacterium* phage FP_oengus were the predominant viruses in feces of healthy piglets, whereas *Escherichia* phage FV3, crAssphage cr116_1, and crAssphage cr8_1 were the predominant viruses in feces of piglets with diarrhea (Fig. 5F).

For RNA viruses at the phylum level, *Pisuviricota*, *Uroviricota*, and *Duplornaviricota* were the predominant viruses in feces of healthy piglets, whereas *Pisuviricota*, *Dividoviricota*, and *Uroviricota* were the predominant viruses in feces of piglets with diarrhea (Fig. 6A). At the class level, *Stelpaviricetes*, *Pisoniviricetes*, and *Caudoviricetes* were the predominant viruses in feces of healthy piglets, whereas *Stelpaviricetes*, *Pisoniviricetes*, and *Laserviricetes* were the predominant viruses in feces of piglets with diarrhea (Fig. 6B). At the order level, *Stellavirales*, *Picornavirales*, and *Caudovirales* were the predominant viruses in feces of healthy piglets, whereas *Stellavirales*, *Picornavirales*, and *Halopanivirales* were the predominant viruses in feces of piglets with diarrhea (Fig. 6C). At the family level, *Astroviridae*, *Caliciviridae*, and *Picornaviridae* were the predominant viruses in feces of healthy piglets, whereas *Astroviridae*, *Picornaviridae*, and *Caliciviridae* were the predominant viruses in feces of piglets with diarrhea (Fig. 6D). At the genus level, *Mamastrovirus*, *Sapovirus*, and *Enterovirus* were the predominant viruses in feces of healthy piglets, whereas *Mamastrovirus*, *Enterovirus*, and *Sapovirus* were the predominant viruses in feces of piglets with diarrhea (Fig. 6E). At the species level, porcine astrovirus 2, Sapporo virus, and astrovirus wild boar/WBAstV-1/2011/HUN were the predominant viruses in feces of healthy piglets, whereas porcine astrovirus 2, astrovirus wild boar/WBAstV-1/2011/HUN, and porcine astrovirus 4 were the predominant viruses in feces of piglets with diarrhea (Fig. 6F).

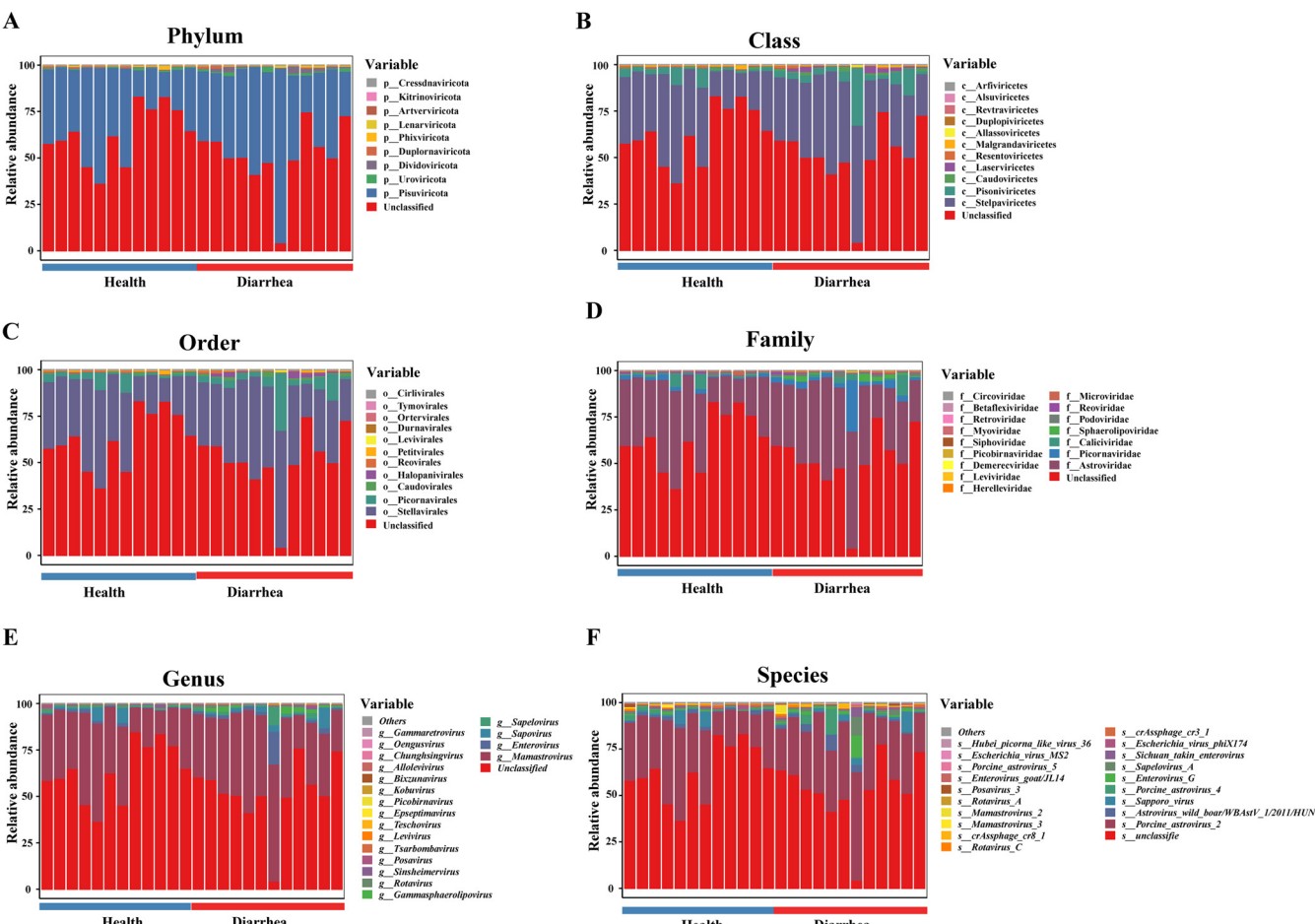

**FIG 6** Composition of intestinal RNA viruses in early-weaned healthy piglets and piglets with diarrhea at the phylum (A), class (B), order (C), family (D), genus (E), and species (F) levels.

**Differential analysis of fecal DNA and RNA viral communities in early-weaned healthy piglets and piglets with diarrhea.** We identified a total of 2, 2, 14, and 40 discriminative DNA viruses between the diarrhea and healthy groups at the phylum, family, genus, and species levels, respectively (see Table S1 in the supplemental material). Compared with healthy piglets, piglets with diarrhea were characterized by 2 enriched viruses at the phylum level (Fig. 7A). Compared with healthy piglets, piglets with diarrhea were characterized by 2 enriched viruses at the family level (Fig. 7B). Compared with healthy piglets, piglets with diarrhea were characterized by 9 enriched viruses and 5 depleted viruses at the genus level (Fig. 7C). Compared with healthy piglets, piglets with diarrhea were characterized by 32 enriched viruses and 8 depleted viruses at the species level (Fig. 7D).

We also identified a total of 2, 2, 2, 3, 3, and 4 discriminative RNA viruses between the diarrhea and healthy groups at the phylum, class, order, family, genus, and species levels, respectively (see Table S2). Compared with healthy piglets, piglets with diarrhea were characterized by 1 enriched virus and 1 depleted virus at the phylum level (Fig. 8A). Compared with healthy piglets, piglets with diarrhea were characterized by 1 enriched virus and 1 depleted virus at the class level (Fig. 8B). Compared with healthy piglets, piglets with diarrhea were characterized by 1 enriched virus and 1 depleted virus at the order level (Fig. 8C). Compared with healthy piglets, piglets with diarrhea were characterized by 2 enriched viruses and 1 depleted virus at the family level (Fig. 8D). Compared with healthy piglets, piglets with diarrhea were characterized by 2 enriched viruses and 1 depleted virus at the genus level (Fig. 8E). Compared with healthy piglets, piglets with diarrhea were characterized by 2 enriched viruses and 2 depleted viruses at the species level (Fig. 8F).

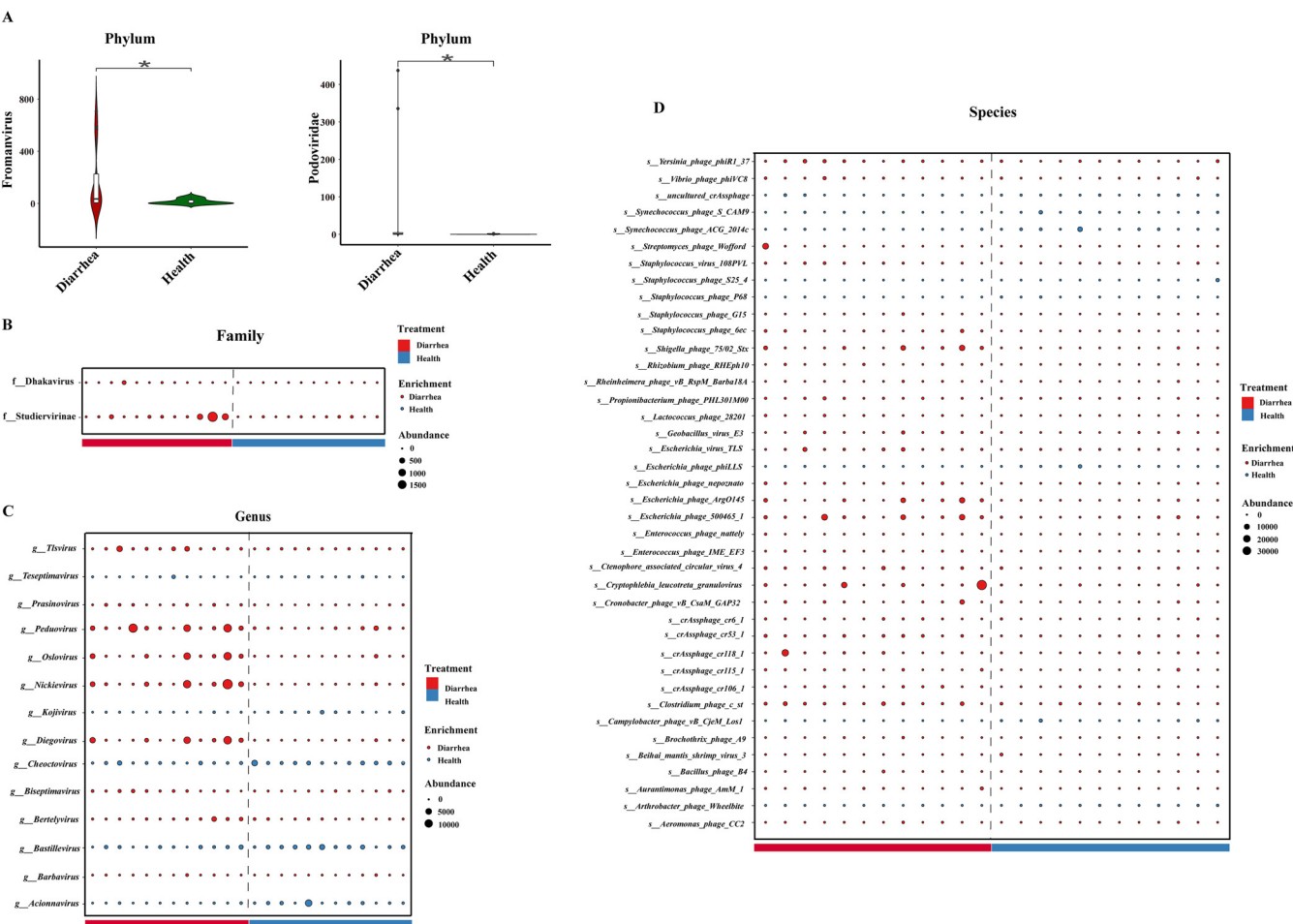

**FIG 7** Intestinal differential DNA viruses in early-weaned healthy piglets and piglets with diarrhea at the phylum (A), family (B), genus (C), and species (D) levels. *, $P < 0.05$ ($n = 12$).

**Cooccurrence analysis of DNA and RNA viruses.** We next explored the potential correlations of abundances of these differential gut DNA and RNA viruses (see Tables S3 to 5S). Overall, the cooccurrence analysis showed that DNA and RNA formed strong and extensive cooccurrence relationships with each other. Within this coexpression network, DNA and RNA viruses had 32 and 2 correlations, respectively, with themselves. In addition, there were 10 correlations between DNA and RNA viruses. The 32 correlations between DNA viruses themselves were all positive (Fig. 9A). One positive correlation and 1 negative correlation were formed between RNA viruses themselves (Fig. 9B). Eight positive correlations and 2 negative correlations were formed between DNA and RNA viruses (Fig. 9C).

## DISCUSSION

In recent years, researchers have gradually begun to focus on the characterization of human and animal gut viromes. A recent study used an assembly metagenomic approach to construct a reference database of DNA viruses from human intestinal microbiomes (23). Another study used a high-throughput metatranscriptomic approach to analyze the viromes of 1,941 game animals (5 orders and 18 species) from 20 provinces in China (24). These studies have greatly expanded the scientific community's knowledge of the mammalian virome. Although the enteric virome is generally considered to be an important component of the intestinal microorganisms, relatively few studies have been conducted on the composition of the enteric virome in different disease states (25, 26). Emerging studies have shown that the enteric viral community can modulate bacterial flora

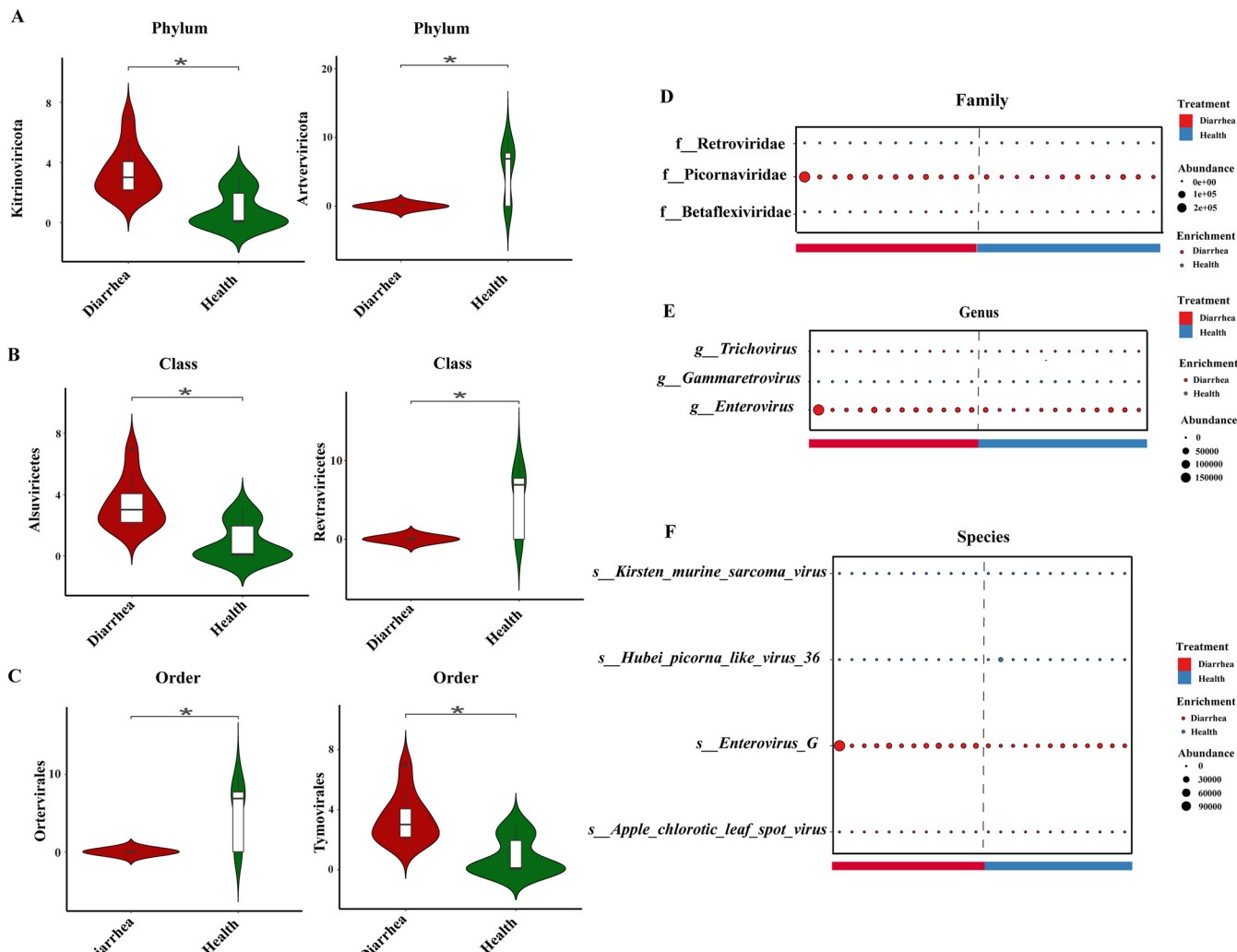

**FIG 8** Intestinal differential RNA viruses in early-weaned healthy piglets and piglets with diarrhea at the phylum (A), class (B), order (C), family (D), genus (E), and species (F) levels. *, $P < 0.05$ ($n = 12$).

composition, profoundly affecting host physiology and promoting the development of intestinal diseases such as inflammatory bowel disease (IBD) and colorectal cancer (27–29). Early-weaning-induced stress in piglets is usually accompanied by intestinal inflammation and impairment of intestinal barrier function, which seriously endanger animal husbandry (30, 31). In the present study, we systematically analyzed the structure and composition of the enteric virome in early-weaned healthy piglets and piglets with diarrhea at all biological classification levels (phylum, class, order, family, genus, and species).

The colonization of the intestinal microbiota of piglets starts immediately after birth; thereafter, as the intestinal luminal environment changes and early weaning occurs, the colonization sites of specific microbiota in the intestine shift and the structure of the microbiota changes (32, 33). Early-weaning-induced stress has a profound effect on the intestinal microbiota of piglets. It is generally accepted that the abnormal intestinal microbiota structure in weaned piglets with diarrhea is mainly characterized by a decrease in diversity, an increase in the relative abundance of pathogenic bacteria, and a decrease in the relative abundance of commensal bacteria (34–36). To date, however, no study has examined whether early-weaned piglets are accompanied by disorders of the enterovirus community. Our data showed that the richness index values of DNA viruses at the family and species levels were significantly higher in early-weaned piglets with diarrhea than in healthy piglets. For RNA viruses, the richness

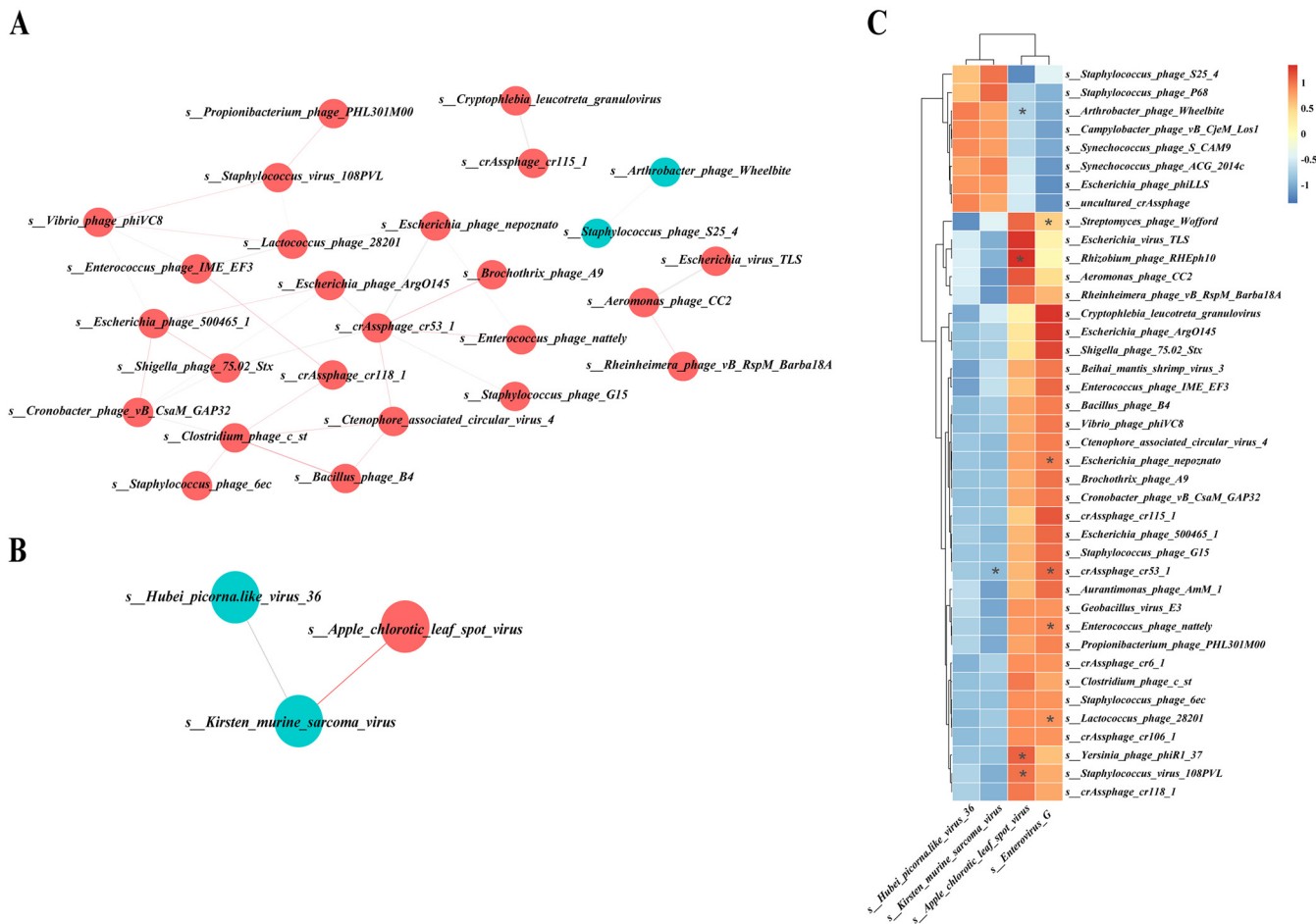

**FIG 9** Cooccurrence network or heatmap constructed from the relative abundances of differential DNA and RNA viruses in early-weaned healthy piglets and piglets with diarrhea at the species level. (A) Correlation between DNA and DNA viruses. (B) Correlation between RNA and RNA viruses. (C) Correlation between DNA and RNA viruses.

index at the genus and species levels, the Shannon index at the phylum, order, family, and genus levels, and the Simpson index at the phylum and family levels were significantly higher in the early-weaned piglets with diarrhea. In addition, the PCoA structures of gut viromes in early-weaned healthy piglets and piglets with diarrhea were similar at all biological classification levels. These results suggested that, although the overall structures of the gut virome are similar in weaned piglets with different healthy statuses, piglets with diarrhea may have a higher number of virus species and greater diversity of virus communities, compared to healthy piglets.

Pig viruses have been investigated for about 100 years, using traditional isolation or PCR methods. A recent study described virus populations from 1,841 healthy weaned piglets from 45 commercial farms in 25 major pig-producing regions across China and created a virus sequence data set called Pigs_VIRES, which matched 96,586 virus genes from at least 249 genera in 66 families, almost 3 times the number of previously published porcine virus genes (37). Unfortunately, that study did not include information on the virome of early-weaned piglets with diarrhea. In this study, we revealed the species composition of DNA and RNA viruses in early-weaned healthy piglets and piglets with diarrhea. Our data showed that the species compositions of intestinal DNA and RNA viruses in early-weaned healthy piglets and piglets with diarrhea are almost identical at all biological taxonomic levels, differing only in the relative abundances of some viruses. These results suggested that early-weaning-induced stress has little effect on species members of the piglet enteric virome and that imbalances in the abundance of certain specific viruses may be key biological markers to discriminate between healthy piglets and piglets with diarrhea.

Results of a recent study reported that the enteric virome is dysregulated in patients with IBD and induces an inflammatory innate immune response in patients with IBD, compared to the enteric virome-induced anti-inflammatory innate immune response in non-IBD individuals (38). In addition, the non-IBD enteric virome was in remission, while the IBD enteric virome worsened dextran sulfate sodium (DSS)-induced colitis in mice. The results of that study suggest that the enteric virome can affect intestinal homeostasis and IBD by modulating the innate immune response (38). Early-weaning-induced stress has been widely demonstrated to cause intestinal inflammatory responses in piglets (39–41). A previous study provided preliminary insights into the relationship between the enteric virome and diarrhea in neonatal piglets of commercial production systems (42). However, the role of the gut virome in early-weaned piglets with diarrhea has not been explored. Here, we identified 40 DNA viruses and 4 RNA viruses at the species level that were involved in early-weaned piglets with diarrhea. Among these differential virus species, 80% of DNA viruses and 50% of RNA viruses were significantly enriched in early-weaned piglets with diarrhea. Notably, we found the family *Dhakavirus* (DNA virus) in the early-weaned healthy piglets, while the phylum *Artverviricota*, the class *Revtraviricetes*, the order *Ortervirales*, the family *Retroviridae*, the genus *Gammaretrovirus*, and the species Kirsten murine sarcoma virus (RNA viruses) were absent in the piglets with diarrhea. These findings indicate that there is value in exploring the role of these enteroviruses in the development of diarrhea in early-weaned piglets. In addition, we found a strong correlation between some DNA and RNA viruses, suggesting that these viruses may act through synergistic or antagonistic forms. However, these findings remain preliminary and still need to be further explored.

In conclusion, to the best of our knowledge, our study is the first to perform comparative analysis of the characteristics of the enteric virome in early-weaned healthy piglets and piglets with diarrhea by metagenomic and metatranscriptomic techniques. We found that the overall structure and species members of the gut virome were similar in piglets with different health condition after early weaning. However, piglets with diarrhea generally possessed greater ecological diversity of intestinal DNA and RNA viruses than did healthy piglets. We also identified differences in the gut virome of healthy piglets and piglets with diarrhea at different biological taxonomic levels. It is noteworthy that some specific viruses were completely absent from the gut of healthy piglets or piglets with diarrhea, suggesting their potential as biomarkers for the prediction or diagnosis of diarrhea in early-weaned piglets. However, the role of the gut virome in early-weaning-induced stress leading to piglet diarrhea needs more investigation. Nonetheless, we think that our findings make a corresponding contribution to revealing the characteristics of the gut virome in early-weaned piglets.

## MATERIALS AND METHODS

**Animal management and sample collection.** A total of 200 piglets (Duroc × Landrace × Yorkshire) was used for selection of healthy piglets and piglets with diarrhea after weaning. All sows were fed the same diets during gestation and lactation, and the piglets were weaned at 21 days of lactation. On the morning of the day of weaning (21 days of age), all piglets were transferred to the nursing room. These weaned piglets were fed a corn/soybean-based diet and had free access to water and feed. All nutrients reached or exceeded National Research Council (NRC) 2012 recommendations for piglets. The general health of each piglet was closely monitored after weaning, and special attention was paid to fecal consistency. According to these observations, a piglet with diarrhea or a healthy piglet was defined based on the criteria described in a previous study (43). Briefly, piglets with diarrhea were characterized as those suffering from diarrhea for at least 2 consecutive days with liquid and watery feces that had not received antibiotic therapy prior to sample collection; meanwhile, healthy piglets were those that had never experienced diarrhea or other diseases. Finally, we randomly selected 12 healthy piglets and 12 piglets with diarrhea and collected the feces of these piglets. All collected fecal samples were immediately frozen in liquid nitrogen and stored at −80℃. All of the animal experimental procedures and sample collection procedures were approved by the Institutional Animal Care and Use Committee of the Huazhong Agricultural University (Hubei, China). In this study, all experimental methods were performed in accordance with the Huazhong Agricultural University of Health Guide for the Care and Use of Laboratory Animals.

**DNA extraction, library construction, and metagenomic sequencing.** Total DNA was extracted from fecal samples using the E.Z.N.A. DNA kit (Omega Bio-tek, Norcross, GA) according to the

manufacturer's protocols. High-quality DNA samples (optical density at 260 nm [$OD_{260}$]/$OD_{280}$ of 1.8 to 2.2 and $OD_{260}$/$OD_{230}$ of ≥2.0) were used to construct sequencing libraries. Metagenomic libraries were prepared following the TruSeq Nano DNA sample preparation kit from Illumina (San Diego, CA), using 1 μg of total DNA. DNA end repair, A-base addition, and ligation of the Illumina-indexed adaptors were performed according to the Illumina protocol. Libraries were then size selected for DNA target fragments of ~400 bp on 2% low-range ultra-agarose followed by PCR amplification using Phusion DNA polymerase (New England Biolabs) for 15 PCR cycles. Metagenomic sequencing was performed by Shanghai Biozeron Biotechnology Co., Ltd. (Shanghai, China). All samples were sequenced on the next-generation sequencing platform in paired-end 150-bp read mode.

**RNA extraction, library construction, and metatranscriptomic sequencing.** Total RNA was extracted from the fecal samples using TRIzol reagent (Invitrogen) according to the manufacturer's instructions, and genomic DNA was removed using DNase I (TaKaRa). Then RNA quality was determined using a 2100 Bioanalyzer (Agilent) and quantified using a NanoDrop 2000 spectrophotometer (Thermo Fisher Scientific). High-quality RNA samples ($OD_{260}$/$OD_{280}$ of 1.8 to 2.2 and $OD_{260}$/$OD_{230}$ of ≥2.0) were used to construct sequencing libraries. Metatranscriptomic libraries were prepared using the TruSeq RNA sample preparation kit from Illumina, using 5 μg of total RNA. rRNA removal with Ribo-Zero rRNA removal kits (Epicenter), fragmentation using fragmentation buffer, cDNA synthesis, end repair, A-base addition, and ligation of the Illumina-indexed adaptors were performed according to the Illumina protocol. Libraries were then size selected for cDNA target fragments of 200 to 300 bp on 2% low-range ultra-agarose followed by PCR amplification using Phusion DNA polymerase (New England Biolabs) for 15 PCR cycles. Metatranscriptomic sequencing was performed by Shanghai Biozeron Biotechnology Co., Ltd. All samples were sequenced with the next-generation sequencing platform in paired-end 150-bp read mode.

**Species composition, alpha diversity, PCoA, and differential species analysis of viruses.** First, metaSPAdes was employed to assembly both DNA and RNA reads due to its performance in sensitivity. Following assembly, contigs longer than 10,000 bp was retained for DNA reads. For contigs assembled from RNA reads, 2,000 bp was set as the threshold to filter out short contigs. These contigs were analyzed by VirSorter and VirFinder for viral identification. Contigs sorted as VirSorter category 1, 2, 4, or 5 and those with VirFinder scores of > 0.9 ($P < 0.01$) were considered viral and retained for downstream analysis. CD-HIT (version 4.6.1) was used to dereplicate viral contigs with the options –c 0.95, –G 0, and –aS 0.75 to establish a nonredundant viral catalog. Salmon (version 2.21) was used to calculate the abundance of each viral contig by mapping high-quality reads to nonredundant viral genomes. Taxonomic classification of viral catalog was performed using Kraken2. Alpha diversity analyses, including richness, Shannon, and Simpson index determinations were conducted and visualized using the tidyverse (version 1.3.1), vegan (version 2.5-7), and ade4 (version 1.7-17) packages in R, respectively. PCoA was used to visually evaluate the overall difference and similarity of viral communities between the diarrhea and healthy groups, and permutational multivariate analysis of variance (PERMANOVA) was used to calculate statistical significance between groups (44). The viral species with differences between the two groups were identified using the Wilcoxon rank sum test, and $P$ values of <0.05 were considered statistically significant. The numbers of reads used for metagenomic and metatranscriptomic analyses are presented in Table S6 in the supplemental material.

**Construction of the cooccurrence network.** The cooccurrence network was calculated on the basis of relative abundances with Spearman's rank correlation coefficient ($P < 0.05$) using the R package Hmisc (version 4.5-0). The network layout was calculated and visualized using a circular layout with the Cytoscape software. Only edges with correlations of >0.5 were shown in the two nodes, and unconnected nodes were omitted. Correlation coefficients with magnitudes of ≥0.5 were selected for visualization in Cytoscape (version 3.8.2) (45).

**Data availability.** The data sets supporting the conclusions of this article are available in the NCBI Sequence Read Archive (SRA) repository under BioProject accession number PRJNA775062.

## SUPPLEMENTAL MATERIAL

Supplemental material is available online only.
**SUPPLEMENTAL FILE 1**, PDF file, 0.1 MB.

## ACKNOWLEDGMENTS

We sincerely thank Zhenyu Wang of the China Agricultural University for his technical support in the process of data analysis.

This work was supported by the National Nature Science Foundation of China (grant 31902189), the Natural Science Foundation of Hubei Province (grants 2021CFB436 and 2021CFA018), the Knowledge Innovation Program of the Wuhan-Shuguang Project (grant 2022020801020230), and the Fundamental Research Funds for the Central Universities (grants 2662020DKQD004 and 2662022YJ003).

S.T. designed the experiments and wrote the original draft. H.Z. performed visualization and wrote the original draft. J.L. edited the manuscript. H.Z. and J.L. participated in experiments and sample collection. S.T. and H.W. conceived the study and reviewed the manuscript.

We declare that we have no conflicts of interest.

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
