## [Reviewer comments · Microbiology Spectrum]

Microbiology Spectrum

Landscapes of enteric virome signatures in early-weaned piglets

Shiyu Tao, Huicong Zou, Jingjing Li, and Hong Wei

Corresponding Author(s): Shiyu Tao, Huazhong Agricultural University

Review Timeline:

Submission Date:	May 6, 2022
Editorial Decision:	June 2, 2022
Revision Received:	July 6, 2022
Accepted:	July 11, 2022

Editor: Jinxin Liu

Reviewer(s): Disclosure of reviewer identity is with reference to reviewer comments included in decision letter(s). The following individuals involved in review of your submission have agreed to reveal their identity: Kanchan Bhardwaj (Reviewer #1); Tasha Marie Santiago-Rodriguez (Reviewer #2)

Transaction Report:

DOI: <https://doi.org/10.1128/spectrum.01698-22>

June 2, 2022

Dr. Shiyu Tao
Huazhong Agricultural University
No.1 Shizishan Street, Hongshan District, Wuhan City, Hubei Province, China
Wuhan, Hubei 430070
China

Re: Spectrum01698-22 (Landscapes of enteric virome signatures in early-weaned piglets)

Dear Dr. Shiyu Tao:

Link Not Available

Sincerely,

Jinxin Liu

Journals Department
Reviewer comments:

Reviewer #1 (Public repository details (Required)):

Authors have mentioned that the Dataset has been submitted to NCBI SRA repository and accession number is provided (PRJNA775062). However, it is not accessible.

Reviewer #1 (Comments for the Author):

In this study, authors have performed analysis of the intestinal virome in diarrheic- and healthy early-weaned piglets, through

metagenomics and metatranscriptomics. DNA- and RNA- viruses are analyzed by sequencing of total DNA and total RNA present in the fecal samples of 12 healthy- and 12 diarrheic early-weaned piglets. Based on this analysis, authors report that the alpha-diversity is elevated in the diarrheic piglets but no significant change is observed in the beta-diversity. 8 viruses are identified as differentially present species in the two populations. Also, that this is a first report on comparison of virome composition in healthy and diarrheic early-weaned piglets.

I have following comments-

Major Points:

1. Authors have mentioned that their dataset has been submitted to NCBI SRA repository with accession number PRJNA775062. However, it is not accessible.
2. Authors have reported that two viruses, Dhakavirus and Lactobacillus prophage Lj928 are absent in the healthy (early-weaned) piglets and that six viruses are absent in the diarrheic piglets. However, to be able to consider that these differences are statistically significant, the sample size (12 in each population) used in this study is not sufficient. PCR method could be used to detect the 8 viruses in question in larger sample size to establish the significance.
3. Line 358-359: contigs longer than 10, 000 bp were retained for DNA and the threshold for RNA reads was 2000 bp. The reason for these cut-offs should be mentioned because many DNA virus genomes are shorter than 10, 000 bp.
4. Line 361: Why were VirSorter categories, 3 & 6 excluded?
5. Line 366: What is meant by "viral catalog"?
6. What database(s) is used for taxonomic annotation?

Minor Points:

The authors have called their study population as "health piglets" and "diarrhea piglets" at some places whereas it is mentioned as "healthy piglets" and "diarrheic piglets" at some. Please correct.

Line 76-77: Gut virome cannot be called a technology.

Line 105: Figure "1A-C and E" should be "1A, C and E".

Line 111: "Different at the phylum, class, order and genus" should be changed to "Different at the phylum, class, order and family".

Line 113: "at the phylum, order, family" should be "at the phylum, class, order, family".

Line 199: Please clarify if the co-occurrence analysis is done in healthy, diarrheic or both groups of piglets. Also, is there a rationale to analyze co-occurrence of DNA and RNA viruses?

Line 583: "Species composition of intestinal", should be changed to "Composition of intestinal" because the figure is showing composition at levels other than Species too.

Reviewer #2 (Public repository details (Required)):

Metagenomics and metatranscriptomics data were generated for this study, so they need to be deposited

Reviewer #2 (Comments for the Author):

Major comments:

There are DNA viruses listed as RNA viruses, because they were detected using metatranscriptomics (Supplementary table 2, etc). Can the authors clarify throughout the manuscript that some DNA viruses were detected using metatranscriptomics means that these were actively transcribing genes? For example, in lines 186 and 241, were these RNA viruses based on their genomes, or some of these were DNA viruses that were identified using the metatranscriptomics pipeline and thus were grouped with the RNA viromes? This seems to be the case for some of the Lactobacillus phages and prophages, shown in figure 8, which are DNA viruses, but were captured using metatranscriptomics.

Did the authors look into correlations between the DNA viromes, and the bacteriome? This may be important to discuss and show any bacteria-virus interactions, similar to those shown in Figure 9 between the DNA and RNA viromes.

Figure 1 shows important information regarding alpha diversity, but the figure may be too crowded and deviate from the main message. The figure may benefit from just showing the plots with significant differences, including panels D and F. Alternatively, panels D, E and F can be shown. Panels A, B, and C can be mentioned in the text, or moved to supplementary material.

Same with Figure 2. Alternatively, alpha diversity indices for DNA and metatranscriptomics can be shown in the same figure for both healthy and piglets with diarrhea. This may facilitate comparing the alpha diversity values for both the DNA and RNA viromes in one plot.

The authors mentioned a package called Salmon. Can they further explain how abundances were obtained using this package and if the relative abundances were obtained by dividing the number of contigs by the total number of contigs?

Other comments:

Lines 174-176: Are these numbers respective to the taxonomic levels? Please clarify.

Can the authors clarify the number of reads or sequencing depth used in the analyses? This is important to include as sequencing depth is known to affect virome results.

Staff Comments:

Preparing Revision Guidelines

Please return the manuscript within 60 days; if you cannot complete the modification within this time period, please contact me. If you do not wish to modify the manuscript and prefer to submit it to another journal, please notify me of your decision immediately so that the manuscript may be formally withdrawn from consideration by Microbiology Spectrum.

Weaning of piglets at 21 days after birth is considered early-weaning but it is practiced in pig farms, in order to improve farming efficiency. It has been reported that early-weaned piglets are at risk for intestinal dysfunction, which can lead to diarrhea and even death. In this study, authors have explored if gut virome is associated with diarrhea in early-weaned piglets. They have performed analysis of DNA- and RNA-viruses in diarrheic- and healthy early-weaned piglets, through metagenomics and metatranscriptomics. Total DNA and total RNA, present in the fecal samples of 12 healthy- and 12 diarrheic early-weaned piglets are sequenced and analyzed in this study. Based on this analysis, authors report that (i) the α -diversity is elevated in the diarrheic piglets but no significant change is observed in their β -diversity; (ii) 8 viruses are identified as differentially present species in the two populations; (iii) this is a first report on comparison of virome composition in healthy and diarrheic early-weaned piglets.

I have following comments-

Major Points:

1. Authors have mentioned that their dataset has been submitted to NCBI SRA repository with accession number PRJNA775062. However, it is not accessible.
2. It would be useful to provide Read and Contig Summary for each sample as a suppl. Table.
3. Authors have reported that two viruses, Dhakavirus and *Lactobacillus prophage Lj928* are absent in the healthy (early-weaned) piglets and that six viruses are absent in the diarrheic piglets. However, to be able to consider that these differences are statistically significant, the sample size (12 in each population) used in this study is not sufficient. PCR method could be used to detect the 8 viruses in question in larger sample size to establish the significance.
4. Line 358-359: contigs longer than 10, 000 bp were retained for DNA and the threshold for RNA reads was 2000 bp. The reason for these cut-offs should be mentioned because many DNA virus genomes are shorter than 10, 000 bp.
5. Line 361: Why were VirSorter categories, 3 & 6 excluded?
6. Line 366: What is meant by “viral catalog”?
7. What reference database(s) is used for taxonomic annotation is not clear?

Minor Points:

Authors have called their study populations as “health piglets” and “diarrhea piglets”, at some places whereas they are mentioned as “healthy piglets” and “diarrheic piglets” at some. Please correct.

It would be helpful to mention the extent of early-weaning practice and how it is being managed presently.

What kind of associations have been reported between early-weaning and microbiome of piglets, in earlier studies?

Line 76-77: Gut virome cannot be called a technology.

Line 105: Figure “1A-C and E” should be “1A, C and E”.

Line 111: “Different at the phylum, class, order and genus” should be changed to “Different at the phylum, class, order and family”.

Line 113: “at the phylum, order, family” should be “at the phylum, class, order, family”.

Line 199: Please clarify if the co-occurrence analysis is done in healthy, diarrheic or both groups of piglets. Also, is there a rationale to analyze co-occurrence of DNA and RNA viruses?

Line 583: “Species composition of intestinal”, should be changed to “Composition of intestinal” because the figure is showing composition at levels other than Species too.

July 06, 2022

Manuscript No: Spectrum01698-22

Landscapes of enteric virome signatures in early-weaned piglets

Shiyu Tao*, Huicong Zou, Jingjing Li, Hong Wei*

Microbiology Spectrum

Dear Dr. Jinxin Liu

Thank you very much for your editorial letter on Jun 2, 2022, regarding our manuscript Spectrum01698-22. We have carefully considered the editor and reviewers' comments and revised the manuscript accordingly. All the changes made to the manuscript are indicated in red color.

On behalf of my co-authors, I would like to thank you for your support of our work. We look forward to hearing from you soon regarding this resubmission.

Yours sincerely,

Dr. Shiyu Tao

College of Animal Sciences and Technology, Huazhong Agricultural University,
Wuhan, China 430070.

E-mail: sytao@mail.hzau.edu.cn

Reviewer comments:

Reviewer #1 (Public repository details (Required)):

Authors have mentioned that the Dataset has been submitted to NCBI SRA repository and accession number is provided (PRJNA775062). However, it is not accessible.

Reply: We have changed the opening times of the dataset which is now accessible.

Reviewer #1 (Comments for the Author):

In this study, authors have performed analysis of the intestinal virome in diarrheic- and healthy early-weaned piglets, through metagenomics and metatranscriptomics. DNA- and RNA- viruses are analyzed by sequencing of total DNA and total RNA present in the fecal samples of 12 healthy- and 12 diarrheic early-weaned piglets. Based on this analysis, authors report that the alpha-diversity is elevated in the diarrheic piglets but no significant change is observed in the beta-diversity. 8 viruses are identified as differentially present species in the two populations. Also, that this is a first report on comparison of virome composition in healthy and diarrheic early-weaned piglets.

I have following comments-

Major Points:

1. Authors have mentioned that their dataset has been submitted to NCBI SRA repository with accession number PRJNA775062. However, it is not accessible.

Reply: We have changed the opening times of the dataset which is now accessible.

2. Authors have reported that two viruses, Dhakavirus and Lactobacillus prophage Lj928 are absent in the healthy (early-weaned) piglets and that six viruses are absent in the diarrheic piglets. However, to be able to consider that these differences are statistically significant, the sample size (12 in each population) used in this study is not sufficient. PCR method could be used to detect the 8 viruses in question in larger sample size to establish the significance.

Reply: Thank you very much for your professional suggestion. Based on your suggestion, we tried hard to collect more samples, but since the situation of African Swine Fever in China is still very serious and pig farms are under closed management, we could not get more samples.

In fact, like all other differential viruses, these viruses were also differential viruses identified using the statistical method of Wilcoxon Rank-Sum Test. In addition, a number of reported literatures have demonstrated that 12 biological replicates per group are sufficient to be statistically significant.

Reference:

Jun Hu, Libao Ma, Yangfan Nie, Jianwei Chen, Wenyong Zheng, Xinkai Wang, Chunlin Xie, Zilong Zheng, Zhichang Wang, Tao Yang, Min Shi, Lingli Chen, Qiliang Hou, Yaorong Niu, Xiaofan Xu, Yuhua Zhu, Yong Zhang, Hong Wei, Xianghua Yan. (2018). A Microbiota-Derived Bacteriocin Targets the Host to Confer Diarrhea Resistance in Early-Weaned Piglets. *Cell Host Microbe*, 24(6):817-832.e8. doi: 10.1016/j.chom.2018.11.006.

[https://www.cell.com/cell-host-microbe/fulltext/S1931-3128\(18\)30563-8](https://www.cell.com/cell-host-microbe/fulltext/S1931-3128(18)30563-8)

Zhenyu Wang, Yu Bai, Yu Pi, Walter J J Gerrits, Sonja de Vries, Lijun Shang, Shiyu Tao, Shiyi Zhang, Dandan Han, Zhengpeng Zhu, Junjun Wang. (2021). Xylan alleviates dietary fiber deprivation-induced dysbiosis by selectively promoting *Bifidobacterium pseudocatenulatum* in pigs. *Microbiome*, 9(1):227. doi: 10.1186/s40168-021-01175-x.

<https://microbiomejournal.biomedcentral.com/articles/10.1186/s40168-021-01175-x>

3. Line 358-359: contigs longer than 10, 000 bp were retained for DNA and the threshold for RNA reads was 2000 bp. The reason for these cut-offs should be mentioned because many DNA virus genomes are shorter than 10, 000 bp.

Reply: Importantly, current methods for DNA virus sequence identification cannot reliably identify short (< 10 kb) viral sequences (RNA virus with 2 kb cutoff). Usually for DNA viruses identification, at least 10 kb viral contigs were considered as high-confident viruses with MIUViG criteria.

Reference:

Roux, S., Adriaenssens, E. M., Dutilh, B. E., Koonin, E. V., Kropinski, A. M., Krupovic, M., Kuhn, J. H., Lavigne, R., Brister, J. R., Varsani, A., Amid, C., Aziz, R. K., Bordenstein, S. R., Bork, P., Breitbart, M., Cochrane, G. R., Daly, R. A., Desnues, C., Duhaime, M. B., Emerson, J. B., Enault, F., Fuhrman, J. A., Hingamp, P., Hugenholtz, P., Hurwitz, B. L., Ivanova, N. N., Labonté, J. M., Lee, K.-B., Malmstrom, R. R., Martinez-Garcia, M., Mizrachi, I. K., Ogata, H., Páez-Espino, D., Petit, M.-A., Putonti, C., Rattei, T., Reyes, A., Rodriguez-Valera, F., Rosario, K., Schriml, L., Schulz, F., Steward, G. F., Sullivan, M. B., Sunagawa, S., Suttle, C. A., Temperton, B., Tringe, S. G., Thurber, R. V., Webster, N. S., Whiteson, K. L., Wilhelm, S. W., Wommack, K. E., Woyke, T., Wrighton, K. C., Yilmaz, P., Yoshida, T., Young, M. J., Yutin, N., Allen, L. Z., Kyrpides, N. C., & Elie-Fadrosh, E. A. (2018).

Minimum Information about an Uncultivated Virus Genome (MIUViG). *Nature Biotechnology*, 37, 29.
<https://doi.org/10.1038/nbt.4306>

<https://www.nature.com/articles/nbt.4306#supplementary-information>

4. Line 361: Why were VirSorter categories, 3 & 6 excluded?

Reply: In VirSorter output, there are six categories to evaluate the quality of vOTUs.

category 1 “most confident phage”

category 2 “likely confident phage”

category 3 “possible confident phage”

category 4 “most confident prophage”

category 5 “likely confident prophage”

category 6 “possible confident prophage”

Categories 3 & 6 were always discarded in further analysis.

Reference:

Roux, S., Enault, F., Hurwitz, B. L., & Sullivan, M. B. (2015). VirSorter: mining viral signal from microbial genomic data. *PeerJ*, 3, e985. <https://doi.org/10.7717/peerj.985>

Khan Mirzaei, M., Xue, J., Costa, R., Ru, J., Schulz, S., Taranu, Z. E., & Deng, L. (2020). Challenges of Studying the Human Virome - Relevant Emerging Technologies. *Trends Microbiol.* <https://doi.org/10.1016/j.tim.2020.05.021>

Gregory, A. C., Zayed, A. A., Conceicao-Neto, N., Temperton, B., Bolduc, B., Alberti, A., Ardyna, M., Arkhipova, K., Carmichael, M., Cruaud, C., Dimier, C., Dominguez-Huerta, G., Ferland, J., Kandels, S., Liu, Y., Marec, C., Pesant, S., Picheral, M., Pisarev, S., Poulain, J., Tremblay, J. E., Vik, D., Tara Oceans, C., Babin, M., Bowler, C., Culley, A. I., de Vargas, C., Dutilh, B. E., Iudicone, D., Karp-Boss, L., Roux, S., Sunagawa, S., Wincker, P., & Sullivan, M. B. (2019). Marine DNA Viral Macro- and Microdiversity from Pole to Pole. *Cell*, 177(5), 1109-1123 e1114. <https://doi.org/10.1016/j.cell.2019.03.040>

5. Line 366: What is meant by "viral catalog"?

Reply: We have revised this incorrect description to “viral genomes”.

6. What database(s) is used for taxonomic annotation?

Reply: The taxonomic annotation database was Kraken2 viral database (https://benlangmead.github.io/aws-indexes/k2, k2_viral_20210515).

Minor Points:

The authors have called their study population as "health piglets" and "diarrhea piglets" at some places whereas it is mentioned as "healthy piglets" and "diarrheic piglets" at some. Please correct.

Reply: We have unified “health piglets” and “diarrhea piglets” in the manuscript.

Line 76-77: Gut virome cannot be called a technology.

Reply: Revised.

Line 105: Figure "1A-C and E" should be "1A, C and E".

Reply: Revised.

Line 111: "Different at the phylum, class, order and genus" should be changed to "Different at the phylum, class, order and family".

Reply: Revised.

Line 113: "at the phylum, order, family" should be "at the phylum, class, order, family".

Reply: Revised.

Line 199: Please clarify if the co-occurrence analysis is done in healthy, diarrheic or both groups of piglets. Also, is there a rationale to analyze co-occurrence of DNA and RNA viruses?

Reply: What we used for co-occurrence analysis were differential DNA and RNA viruses between health and diarrhea piglets. At present, the potential link between DNA and RNA viruses has not been well revealed, and our study conducted a Spearman correlation analysis based on differential DNA and RNA viruses to initially suggest a possible link between them.

Line 583: "Species composition of intestinal", should be changed to "Composition of intestinal" because the figure is showing composition at levels other than Species too.

Reply: Revised.

Reviewer #2 (Public repository details (Required)):

Metagenomics and metatranscriptomics data were generated for this study, so they need to be deposited

We have changed the opening times of the dataset which is now accessible.

Reviewer #2 (Comments for the Author):

Major comments:

There are DNA viruses listed as RNA viruses, because they were detected using metatranscriptomics (Supplementary table 2, etc). Can the authors clarify throughout the manuscript that some DNA viruses were detected using metatranscriptomics means that these were actively transcribing genes? For example, in lines 186 and 241, were these RNA viruses based on their genomes, or some of these were DNA viruses that were identified using the metatranscriptomics pipeline and thus were grouped with the RNA viromes? This seems to be the case for some of the Lactobacillus phages and prophages, shown in figure 8, which are DNA viruses, but were captured using metatranscriptomics.

Reply: Thank you very much for your professional suggestion. Some DNA viruses are indeed inevitably detected in metatranscriptome sequencing. Similarly, some RNA viruses may also be found in metagenome sequencing. In order to avoid sending the wrong message to our readers, we have removed the metatranscriptome captured DNA viruses and metagenome captured RNA viruses from the relevant figures and tables.

Did the authors look into correlations between the DNA viromes, and the bacteriome? This may be important to discuss and show any bacteria-virus interactions, similar to those shown in Figure 9 between the DNA and RNA viromes.

Reply: Thank you very much for your professional suggestion. The correlation analysis between bacteria and DNA viruses is indeed beneficial to show the bacteria-virus interactions. In this study, we focused on the intestinal viral characteristics of health and diarrhea piglets. In addition, we have submitted metagenomics sequencing data (bacteria) to other journals, in order to strictly follow academic ethics and avoid conflicts of interest, so we did not involve any information

concerning bacteria in this study. We hope to have your understanding.

Figure 1 shows important information regarding alpha diversity, but the figure may be too crowded and deviate from the main message. The figure may benefit from just showing the plots with significant differences, including panels D and F. Alternatively, panels D, E and F can be shown. Panels A, B, and C can be mentioned in the text, or moved to supplementary material.

Same with Figure 2. Alternatively, alpha diversity indices for DNA and metatranscriptomics can be shown in the same figure for both healthy and piglets with diarrhea. This may facilitate comparing the alpha diversity values for both the DNA and RNA viromes in one plot.

Reply: Thank you very much for your professional suggestion. We believe that even though some of the results are not significantly different, they are still important information in our study. In order to avoid overcrowding of the figures while preserving all the results information, we recombined the content in Figures 1 and 2.

The authors mentioned a package called Salmon. Can they further explain how abundances were obtained using this package and if the relative abundances were obtained by dividing the number of contigs by the total number of contigs?

Reply: The relative abundances of viral genomes were calculated by Salmon within MetaWRAP package which was widely used in metagenomic analysis. In gene abundance calculation, Salmon not only considers the number of contigs and reads, but also considers the GC bias and length of reads. And the detailed algorithm for Salmon can be tracked in the software website (<https://salmon.readthedocs.io/en/latest/>).

Reference:

Uritskiy, G. V., DiRuggiero, J., & Taylor, J. (2018). MetaWRAP-a flexible pipeline for genome-resolved metagenomic data analysis. *Microbiome*, 6(1), 158. <https://doi.org/10.1186/s40168-018-0541-1>

Patro R, Duggal G, Love MI, Irizarry RA, Kingsford C. Salmon provides fast and bias-aware quantification of transcript expression. *Nat Methods*. 2017 Apr;14(4):417-419. Doi: 10.1038/nmeth.4197. Epub 2017 Mar 6. PMID: 28263959; PMCID: PMC5600148.

Other comments:

Lines 174-176: Are these numbers respective to the taxonomic levels? Please clarify.

Reply: Yes, these numbers respective to the taxonomic levels.

Can the authors clarify the number of reads or sequencing depth used in the analyses?

This is important to include as sequencing depth is known to affect virome results.

Reply: According to your suggestion, we have provided the number (sequencing depth) of reads used in metagenomic and metatranscriptomic analyses in supplemental table 6.

July 11, 2022

Dr. Shiyu Tao
Huazhong Agricultural University
No.1 Shizishan Street, Hongshan District, Wuhan City, Hubei Province, China
Wuhan, Hubei 430070
China

Re: Spectrum01698-22R1 (Landscapes of enteric virome signatures in early-weaned piglets)

Dear Dr. Shiyu Tao:

Your manuscript has been accepted, and I am forwarding it to the ASM Journals Department for publication. You will be notified when your proofs are ready to be viewed.

Sincerely,

Jinxin Liu
Editor, Microbiology Spectrum

I found the response to my concerns satisfactory.